# Information based explanation methods for deep learning agents − with applications on large open-source chess models

## Abstract

With large chess-playing neural network models like AlphaZero contesting the state of the art within the world of computerised chess, two challenges present themselves: The question of how to explain the domain knowledge internalised by such models, and the problem that such models are not made openly available. This work presents the re-implementation of the concept detection methodology applied to AlphaZero in McGrath et al. (2022), by using large, open-source chess models with comparable performance. We obtain results similar to those achieved on AlphaZero, while relying solely on open-source resources. We also present a novel explainable AI (XAI) method, which is guaranteed to highlight exhaustively and exclusively the information used by the explained model. This method generates visual explanations tailored to domains characterised by discrete input spaces, as is the case for chess. Our presented method has the desirable property of controlling the information flow between any input vector and the given model, which in turn provides strict guarantees regarding what information is used by the trained model during inference. We demonstrate the viability of our method by applying it to standard $8 \times 8$ chess, using large open-source chess models.

## 1 Introduction

The methodology for training chess-playing models presented in Silver et al. (2018) constituted a significant departure from the methodology used to develop many of the strongest chess-playing programs. It combined deep neural networks trained using reinforcement learning through self-play with the standard procedure of exhaustive and enumerative search, where the trained neural networks provide a means of learning heuristics used by standard chess-playing programs. However, due to the nature of neural networks, these learned heuristics are notoriously opaque. For chess, this means that it is difficult to know what models with superhuman chess playing abilities, like AlphaZero, have learned. The explainable AI (XAI) method of concept detection (Kim et al., 2018) was applied and discussed in McGrath et al. (2022) for explaining what AlphaZero has learned about chess. While this work presents many interesting insights and avenues for exploring what neural network models learn about chess, it is critically based on a model that is not publicly available. This has several drawbacks. Firstly, this means that the results presented in McGrath et al. (2022) are fundamentally non-reproducible for other researchers, a problem exacerbated by the fact that the computing power necessary to train similar models on standard $8 \times 8$ chess is substantial. Secondly, not having access to the models used in McGrath et al. (2022) means that it is difficult to create and evaluate techniques beyond those already presented. There have been attempts to circumvent this by training models on smaller variants of chess, as discussed in Hammersborg & Strümke (2022), but there is considerably more interest for chess-playing programs as a whole for standard $8 \times 8$ chess. Finally, as the model is not available to the general public, it is difficult to sufficiently showcase the strengths and the subtleties of play generated by such models to the world outside of AI research, for instance the chess community.

Since the creation of AlphaZero, there have been efforts to create open-source variants of the training pipeline presented in Silver et al. (2018) for chess. Among the largest efforts is Leela Chess Zero (Leela Zero Chess Development Community, 2018), a community driven initiative to produce superhuman chess-playing models trained through deep reinforcement learning, Monte Carlo Tree Search and self-play. Models created through such initiatives can serve as "stand-in alternatives" for

the closed-source model used in McGrath et al. (2022). This means that it is possible to use them as a basis for exploring existing and creating new explanation methods in this domain.

While there exist many XAI methods capable of providing explanations for large neural network models, most of them are not directly applicable to chess. We therefore believe that having access to large chess models would greatly facilitate the development of novel explanatory methods for these. While such methods need not be applicable only to chess, providing the ability to both verify and develop such methods directly on large chess models would mean that they could provide great utility for both XAI as a field, but also for chess enthusiasts in general. Based on this, it is desirable to use state of the art chess playing models for developing novel XAI methods specifically addressing the problem of chess, while still being applicable to other problem spaces.

Creating visual explanatory methods for chess requires overcoming the perturbation-based nature of it as a problem space: Removing or adding a single piece on a chess board is likely to drastically alter the nature of any given position, meaning that it is difficult to isolate the "contribution" of any single piece. However, this manner of thinking is still useful, both when evaluating, but also when playing chess. Deciding what pieces of a given position are necessary for correctly assessing, or predicting, in machine learning terms, the best move is a form of explanation that would be intuitively useful for a human observer. If such an explanation is supported by the given chess-playing model not being allowed to observe pieces that the explanation deems unimportant, this would mean that the explanatory method has an inherent correctness: it is guaranteed to be representative of the information that the model uses to make its predictions.

Based on the aforementioned challenges, this work makes the following two contributions:

- Demonstrate that it is possible to replicate the results presented in McGrath et al. (2022) using publicly available, open source models.

- Develop a method for generating visual explanations that provide guaranteed complete enumerations of which pieces of information a given model uses to make its predictions, and that the module for generating these explanations is trainable and interlinked with the model itself.

## 2 BACKGROUND

### 2.1 LEELA CHESS ZERO

Leela Chess Zero is an open source, community initiative for training superhuman chess models through self-play, in the same style as AlphaZero (Silver et al., 2018). Resulting models have proven to be capable chess players. The project solves the main bottleneck of training AlphaZero-like chess agents, namely generating training data by self-play, by generating such self-play games through community-wide distributed computing.[1] This circumvents the need for large, co-located clusters of computing power, since it only requires the maintenance of a central server for receiving self-play games, and fitting the models to the gathered data.

The Leela Chess Zero project has produced many iterations of strong chess playing models. The project has also evaluated model architectures beyond those discussed in McGrath et al. (2022), producing models that are believed to be stronger than the models presented in Silver et al. (2018) and McGrath et al. (2022). The project also produced a family of models[2] using the same architecture as AlphaZero presented in Silver et al. (2018). The fully-trained models, in addition to checkpoints taken during training, are made available by the Leela Chess Zero initiative. It is therefore interesting to investigate if the results with regards to explainability and presence of information from this "open-source AlphaZero" are comparable to the results presented in McGrath et al. (2022), produced using AlphaZero.

### 2.2 TRAINING CHESS MODELS WITH MONTE CARLO TREE SEARCH

Chess models like AlphaZero are trained by a combination of Monte Carlo Tree Search Coulom (2007), and deep reinforcement learning by self-play. For a given position $s$, the models are trained to predict a policy vector $p(s)$, a probability distribution over all possible moves from $s$, and $v(s)$, the predicted outcome from $s$. The procedure generates self-play games by using $p(\cdot)$ and $v(\cdot)$ from

---

[1]This is conceptually similar to initiatives such as `Folding@home` (Larson et al., 2009).
[2]Numbered as model(s) `T30`

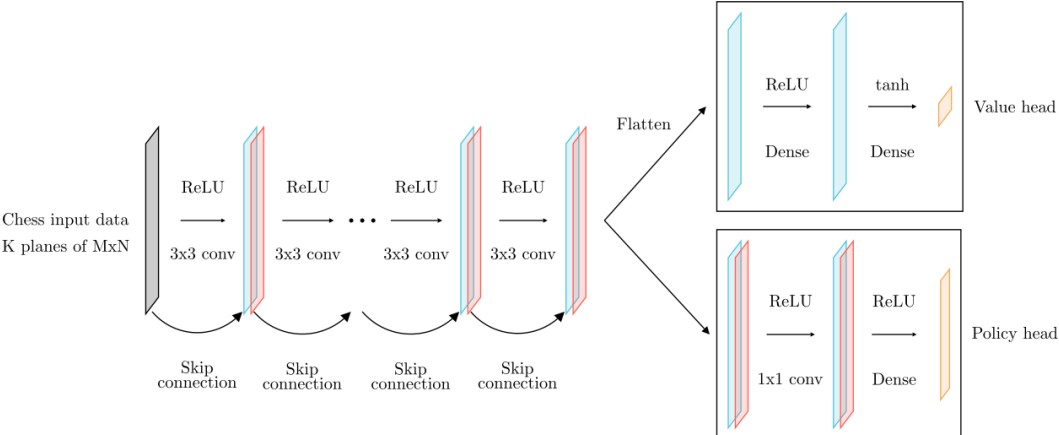

Figure 1: The architecture of the described AlphaZero-like model. The structure is the same as described in McGrath et al. (2022).

the current model iteration to guide a Monte Carlo Tree Search (MCTS) procedure. This means that the current model iteration is implicitly responsible for choosing which moves are selected in each such self-play game. However, as discussed in Silver et al. (2018), this also means that the produced games are of higher quality than if no such search procedure was to be used.

## 2.3 MODEL ARCHITECTURE

The architecture of the models used for this work is almost identical to those used in McGrath et al. (2022). A singular chess position is encoded as a $(8, 8, 21)$-dimensional tensor, where each such channel contains a $(8, 8)$ "plane" of relevant information from the position. As described in McGrath et al. (2022), the first 12 input planes are binary maps separately representing the presence or absence of each type of piece for each player. Additionally, the remaining planes encode auxiliary information about the position, such as which player is the player to move, castling rights for both players, and the total number of turns taken thus far.

For a given position, the models also require the $h - 1$ preceding positions, meaning that a complete training sample has a dimensionality of $(h, 8, 8, 21)$.[3] For the models used in this work, $h = 8$. The models are implemented as standard residual networks He et al. (2015), and in this case consist of 20 residual blocks. This is illustrated in Fig. 1.

## 3 METHOD

### 3.1 REPLICATION OF CONCEPT DETECTION

Concept detection, as first described in Kim et al. (2018), is an XAI method that aims to investigate what information a given neural network model learns to represent in the course of training. More specifically, this method aims to provide a way of detecting the presence of densely represented concepts in the intermediate space of a neural network model by using logistic probes. Given a model $M : I \rightarrow O$ from a input space $I$ to a output space $O$, with an intermediary model function $L_i(s)$ that produces the intermediate activations of the $i$-th layer in $M$, and a concept function $f(s)$ that describes the presence or absence of a given concept in $s$, one looks to see if there is a simple mapping between $L_i(s)$ and $f(s)$, i.e. if the model learns to find $f(s)$ as a part of its internal representation.

Binary concept detection for the $i$-th layer of $M$ is performed by gathering a set of samples $(L_i(s_j), f(s_j))$ for a given set of states $s_j \in S$. A subset of $S$ is withheld as a validation set.

---

[3]The utility of including previous positions as a part of any such input sample is quite interesting, since only including the current state includes all relevant information for producing an evaluation or move prediction in terms of pure informational content. McGrath et al. (2022) report that this provides an empirical increase in performance. However, this introduces the possibility that a model might produce differing predictions for the same state, solely depending on the positions leading up to it.

Table 1: Concept functions used for concept detection. Adapted for use from Hammersborg (2023).

| Name | Description |
|------|-------------|
| has_mate_threat | Checkmate is available |
| in_check | Is in check |
| material_advantage | Has more pieces than opponent |
| threat_opp_queen | Opponent's queen can be captured |
| has_own_double_pawn | Has two pawns on the same file |
| has_opp_double_pawn | Opponent has two pawns on the same file |
| has_contested_open_file | Both players have rooks in an open file |
| threat_my_queen | Own queen can be captured |
| random | Data set with random labels |

One then aims to fit a logistic probe to approximate the map $L_i(s) \to f(s)$, by minimising

$$\|\sigma\left(\mathbf{w} \cdot L_i(s_j) + \mathbf{b}\right) - f(s_j)\|_2^2 + \lambda\|\mathbf{w}\|_1 + \lambda\,|\mathbf{b}|\,, \tag{1}$$

for each pair $(L_i(s_j), f(s_j))$ from the training set of $S$, where $\mathbf{w}$ and $\mathbf{b}$ are the trainable parameters of the probe, and $\sigma$ is the standard sigmoid function. The detected presence of a concept is then defined to be binary accuracy of the trained probe on the validation set of $S$ corrected for guessing,

$$\frac{2}{N}\left(\sum_j H\left(\mathbf{w} \cdot L_i(s_j) + \mathbf{b} - 0.5\right) - f(s_j)\right) - 1\,, \tag{2}$$

where $H(\cdot)$ is the Heaviside-function.

This work applies concept detection to the Leela Chess Zero models described in Secs. 2.1 and 2.2, with the intention of replicating relevant results from McGrath et al. (2022). Model iterations were chosen such that the amount of iterations between each subsequent model becomes larger as the model progresses.[4] This was done in order to highlight interesting developments in the model's progress early in the training procedure. The concepts used are listed and described in Table 1. $\lambda$ was chosen such that $\lambda \in \{0.01, 0.001\}$, and the reported values are the maximum value for these selections of $\lambda$. This is the same procedure as used in McGrath et al. (2022).

## 3.2 FAITHFUL REPRESENTATION OF INFORMATION USAGE

### 3.2.1 MOTIVATION

While saliency based explanations, (see e.g. Simonyan et al. (2013); Zeiler & Fergus (2014); Springenberg et al. (2014); Sundararajan et al. (2017); Patro et al. (2019)), can be viable for a variety of problem spaces, most existing methods for generating such explanations are not suitable for application to the domain of chess. Perturbation based methods (e.g. ImageSHAP Lundberg & Lee (2017)) are not viable since it is not obvious how to perturb a given state while assuring that the perturbation remains "close" in the model's input space, i.e., the chess board. That is, while a position can be visually similar in terms of the placement of pieces, the nature of any given position is more often than not significantly changed by even a small perturbation, such as moving a single piece on the board.

Similarly, we also found that the usefulness of existing gradient based methods, such as the widely used GradCAM (Selvaraju et al., 2017), is limited for our case of chess. While the model architecture described in Sec. 2.2 allows for direct application of GradCAM, it lacks the features that would make it ideal for a saliency map based explanation approach. We present GradCAM saliency maps applied to a set of positions for the models presented in Sec. 2.2 in Fig. 2, which highlight our main concern, namely that the generated saliency maps do not necessarily apply to the entire position. Here, while the saliency map correctly highlights pieces and squares that are important for the given

---

[4]Since each the amount of games for each iteration varies for the models used in this work, there is no way of creating a direct mapping between these models and the models used in McGrath et al. (2022).

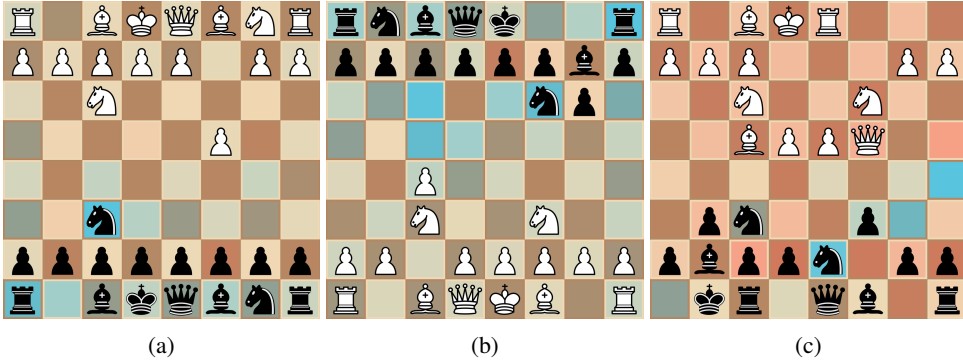

Figure 2: Saliency maps generated for a set of positions for the models described in Sec. 3.2.3 by using GradCAM. We use a color-map that maps lower values towards red, and higher values towards blue.

position, it is safe to assume that the model is not in fact indifferent to the pieces that are not highlighted. Additionally, there have been several inquires challenging the accuracy and dependability of GradCAM, as discussed in Adebayo et al. (2018).

We therefore aim to present an alternative method for generating saliency maps that provide strong guarantees regarding what information the model uses to make its predictions. In broad terms, we achieve this by appending a structure providing direct control over what parts of a given input sample is made available to the model during training. This structure is also trainable, meaning that it learns which parts of a given input sample are relevant for the chess playing model. We call such a saliency map an "information importance map", abbreviated II-map.

### 3.2.2 OVERVIEW

The objective behind the presented method is to allow saliency map generation for neural network models. For a given model $M : I \to O$ as a model from its input space $I$ to its output space $O$, and $s \in O$, the main contribution of our method is to add a trainable reductive operation $R(s)$ as a pre-step for $M$, which constitutes a saliency map over the state $s$.

The intention behind representing a state $s$ by using $R(s)$ is that $R(\cdot)$ should remove information from $s$ that is not relevant when performing inference with the model $M(s)$. Since $R(\cdot)$ is trainable, it can be trained along with $M(\cdot)$, meaning that the information needed by $M(\cdot)$ is accommodated by $R(\cdot)$. A high-level illustration of this is shown in Fig. 3. The apparent benefit of this strategy wrt. other methods for generating saliency maps, is that it provides strong guarantees that the information indicated to be irrelevant by $R(\cdot)$ is never used by $M(\cdot)$. That is, if $R(\cdot)$ has learned that some piece of information from $s$ is irrelevant for the prediction, then $M(\cdot)$ has no way of obtaining it.

### 3.2.3 IMPLEMENTATION

The reductive operation $R(s)$ is implemented as a trainable neural network that produces a stochastic binary mask over a input tensor $s$. This is done by training the model to predict a probability tensor $P$ with a corresponding probability for each element of $s$. The function that produces $P$ given $s$ is the only trainable part of $R(\cdot)$. $P$ is then used to produce a binary mask over $s$ by sampling $X_i \sim \mathcal{U}(0, 1)$ for each element $s_i$ in $s$, evaluated as

$$P_{bin}(s) = H(P(s) - X), \tag{3}$$

where $H$ is the standard Heaviside function. This means that the $i$-th element in $P_{bin}(s)$ is 1 with probability $P_i$. Finally, $R(s)$ is produced by using $P_{bin}(s)$ as a binary mask over $s$. While training, a small L1-penalty is also applied to the sum of each element in $P$, meaning that the training procedure should also seek to minimise the amount of non-zero elements in each produced $P$. The result of training a function to produce a minimal $P$ for a given state $s$, and the usage of $P_{bin}(s)$ to produce $R(s)$ means that $P$ can be used as a saliency map over $s$, where each element in $P$ indicates the importance of the corresponding element in $s$ wrt. the main model $M$. The practical interpretation of $P$ and $P_{bin}(s)$ is that $P$ should contain a probability for each element in $s$, and that each element

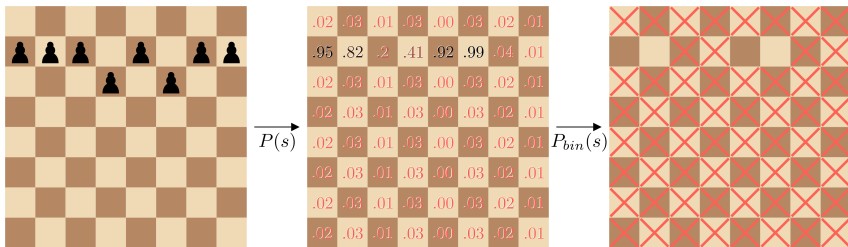

(a) An illustration of the module stochastically producing a mask $H(x)$ for a given position, and the process of binarizing it.

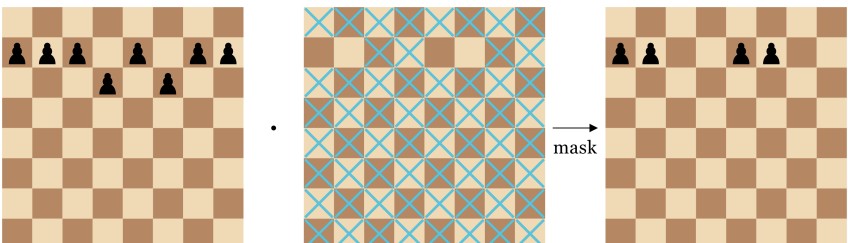

(b) An illustration of how the binary mask produced by the masking module is applied to a given position.

Figure 3: A sample of how the masking module produces a reductive mask during training for a single input channel (a), and how that reductive mask is applied to the given training sample (b).

in $P$ designates the probability that the corresponding element in $s$ is not removed before being passed to the main model $M$.

To make the described method viable for use with backpropagation while training neural networks, a gradient estimation strategy for $R(\cdot)$ is needed. In this case, the only point of contention is estimating $H(\cdot)$, as all other parts of $R(\cdot)$ can be treated as standard operations for neural networks. We use the straight-through estimator, first described in Bengio et al. (2013), for estimating the gradient of $H$, implemented by the `larq`-library (Geiger & Team, 2020) for Python.

While the method provides strong guarantees regarding what information is given to the model, the main drawback is that the reductive model $R(\cdot)$ has to be trained along with $M(\cdot)$. This is fine for smaller models, but for large, chess-playing models such as those described in Sec. 2.2, this poses a challenge. McIlroy-Young et al. (2020) estimates that a chess-playing model trained on actual games requires about $12,000,000$ games to be accurate to an acceptable degree, which is a substantial learning task. We approach this problem by creating a strategy for duplicating a trained model $M_{trained}(\cdot)$ with $R(\cdot)$.[5] Given a trained model $M_{trained}(\cdot)$ that produces a policy vector $p(s)$ over all possible moves from $s$, we first create a dataset $(s, p(s))$ for all states $s$ for all available games in the training set. Then, we train a model $M(\cdot)$ with a reductive step $R(\cdot)$ to predict $p(s)$ from $s$. The main assumption is that the amount of information in $p(s)$ is higher than if one were to train directly on played moves from games directly, since the policy-vector $p(s)$ should consider all moves from $s$, while a training sample from an actual game only presents a single candidate move per position. This means that we can reduce the training task wrt. the number of games required.

We apply $R(\cdot)$ to the first 12 input planes of our model, meaning that we only seek to reduce the information in the planes containing positional information.[6] This means that we only seek to reduce the information in the planes containing positional information. The training of $M(\cdot)$ and $R(\cdot)$ is done through a standard supervised learning procedure. We additionally aim to minimise $\sum P(s)$

---

[5]This is often referred to as "distilling" the model.

[6]The full input structure of our model is described in Appendix A.1.

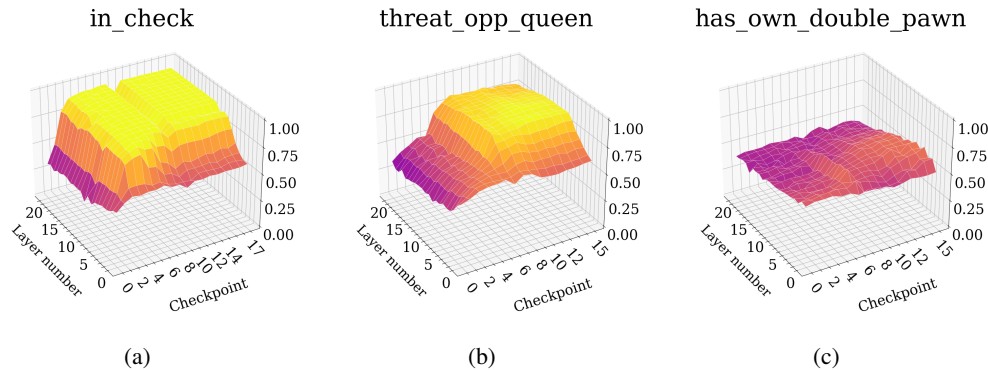

Figure 4: Select concepts modelled by the 8x8 chess agent, showing the model's ability to detect (a) whether the player to move has a material advantage, (b) whether the opponent's queen is under threat, and (c) whether it has a double-pawn. Additional concepts are shown in Appendix A.2.

for all states $s$ in the training set. This is done by adding a standard L1-penalty to $P(s)$ as a term in the loss function used for training $R(\cdot)$ and $M(\cdot)$. When training, the weighting of the L1-penalty can be tweaked to change how important it should be to reduce the amount of information in any given input state. This is usually at the cost of model accuracy, as there is likely to be an inverse relationship between regularisation strength for the masker and prediction performance.[7] $R(\cdot)$ is configured to produce a binary mask with dimensions $(8, 8, 12)$, meaning that the produced mask has one channel per corresponding input channel type, i.e., one channel per combination of piece and colour. Since the produced mask has multiple channels, we devise the following strategy for visualising it. The presented visualisation of a produced mask $P_{bin}(s)$ for state $s$ is given by

$$O_{i,j} = \begin{cases} P_{bin}(s)_{i,j,p} & \text{if square (i, j) occupied by piece type } p \\ \max P_{bin}(s)_{i,j,:}, & \text{otherwise.} \end{cases} \tag{4}$$

In practical terms: if a square is occupied, we show the predicted importance for the type and colour of the piece occupying the square. If a square is not occupied, we show the maximum predicted importance over **all** combinations of piece and colour.

## 4 RESULTS AND DISCUSSION

### 4.1 CONCEPTS

The strategy presented in Sec. 3.1 is applied to the nine concepts listed in Table 1, and we present the binary accuracy corrected for random guessing as specified in Eq. 2 for each residual block for each of the 12 sampled model iterations. Selected results are shown in Fig. 4, and results for all concepts are shown in Appendix A.2.

For all the presented concepts, we see a striking similarity to the corresponding concept-results as presented in McGrath et al. (2022). We see that the model first learns to strongly represent whether it is in check. After this, the model quickly learns to represent threats on its own and the opponent's queen. We additionally observe that pawn-centric concepts are given a stronger representation in earlier layers.

### 4.2 CHESS PUZZLES

We apply our trained model to a set of chess puzzles retrieved from Lichess (2020). Each sample consists of a position, and the first move for the given position that gives a significant advantage for the player to move. These positions are guaranteed to only have a single candidate move to provide the given advantage.

---

[7]It is also worth mentioning that many of the model-masker combinations were very unstable during training. We believe this is mainly caused by the binarisation-procedure discussed in Sec. 3.2.3. Our empirical remedy was reducing the size of the masker, and by reducing the learning rate of our gradient descent procedure.

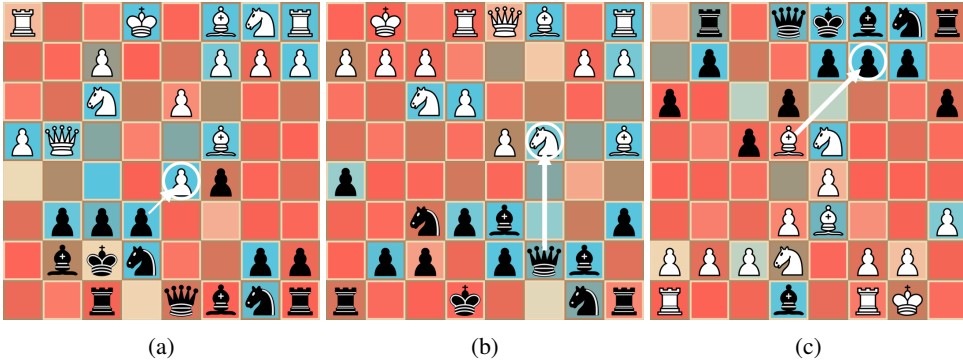

Figure 5: Masks generated for a set of puzzle-positions for the models described in Sec. 3.2.3. We use a color-map that maps lower probability values towards red, and higher probability values towards blue. The move that "solves" the puzzle is shown by a white arrow.

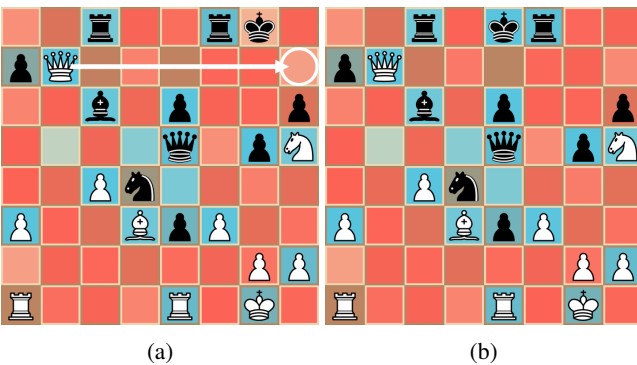

Figure 6: Masks generated for two positions, where (b) is a variation of the position shown in (a) with a more unusual position of the Black king. Observe that the Black king now has larger predicted importance in the more unusual position.

We observe that most of the generated masks for the puzzles shown in Fig. 5 capture the essence of what is necessary to solve the given puzzle. For Fig. 6a, a simple mate-in-one puzzle, we however see that the king (which has to be mated) has a relatively low predicted importance. We hypothesise that this is due to the model predicting it likely for the king to be in the shown position, as combination of rook and king on the upper-right side is a common configuration of pieces. We underpin this by looking at Fig. 6, where we see that moving the king to a less likely square causes it to be predicted with a significantly higher importance.

Additionally, Fig. 5c appears to show the model and masker not finding the optimal move. This is mainly because the bishop that is tasked with carrying out the correct move is predicted with a very low importance, in addition to there being seemingly no other way to infer the presence of this bishop without observing this square directly.

## 4.3 VARIOUS POSITIONS

We also apply our trained model to a set of positions from well known chess games. In contrast to the positions described in Sec.4.2, the main intention is here to observe the produced masks when applying the method to more typical chess situations. We choose three positions from the "Game of the Century", three positions from the first game of the first match between Garry Kasparov and Deep Blue, and three positions from the sixth game in the 2021 World Championship match. Selected positions are shown in Fig. 7.

For the various positions shown in Fig. 7, we see that they capture most immediate threats in almost all positions, (E.g. the threat on Black's knights in Figs. 7a and 7b), in addition to correctly identifying seemingly irrelevant pieces. We also see a mask that resembles being able to verify that the

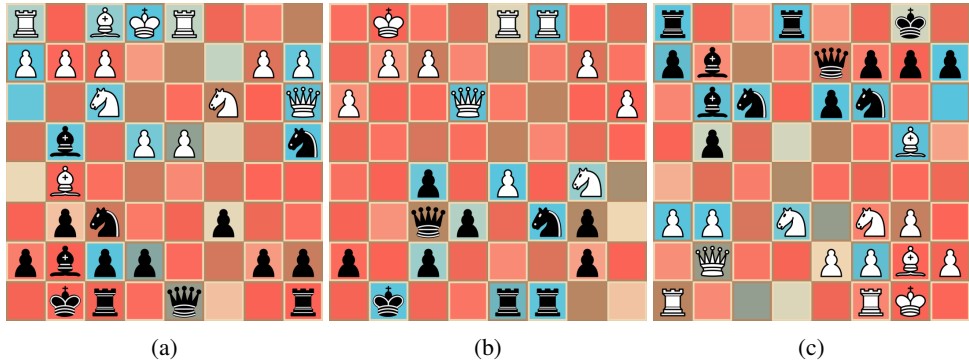

(a)          (b)          (c)

Figure 7: Masks generated for a set of positions from well-known chess games for the models described in Sec. 3.2.3. We use the same color-mapping as used in Fig. 5. The position in Fig. (a) is from "The Game of the Century", Fig. (b) is from the first game between Garry Kasparov and IBM's DeepBlue, and Fig. (c) shows a position from the sixth game from the 2021 World Championships. Additional positions for each game are shown in Appendix A.3.

player to move is in check (Fig. 7c). While the king of the player to move is not masked, we believe this to be the same phenomenon as discussed in Sec. 4.2.

### 4.4 PROPERTIES OF II-MAP

While our proposed method can guarantee the control over the information that reaches the model, its potential as a pure explanatory method can be said to be limited by the fact that it seems to cause the model to learn somewhat adversarial representations. Creating incentives for the model to remove the amount of pieces from the input state might in fact be more a obfuscation of information, rather than the removal of information.

However, this is not to say that the method is without benefit. It is likely that any neural network model, even without the utilisation of an II-map module, learns representations that seem counter-intuitive and adversarial-like. The main benefit of the presented method is in this case to be able to observe and visualise these representations, and in some cases, intepret them, at the computational cost of including an additional masker model in the training loop.

### 5 ONGOING AND FUTURE WORK

The masker model used in this work is available online, with a simple, dynamic user interface that allows for the input of any chess position, available at `https://solid-state-survivor.github.io/ii-map/`. This repository also contains code used for training the models described in Sec. 3.2.3, and is planned to receive updates and further maintenance.[8]

While we have limited our investigations to chess-based models, the II-map method is generally agnostic towards the input space of the model. As an example, our preliminary investigations show that it is possible to train image-based models using a II-map module. However, this training procedure was significantly more unstable than with chess as the input space. Therefore, a significant challenge remains to investigate how to make this training procedure more stable. Additionally, it remains to be seen how large the range of acceptable values for the weight of the L1-penalty used for the masker for all such input spaces, and how the architecture chosen for the masker model affects the rest of the training procedure.

---

[8]The code hosting the user interface and for training our models is contained within an anonymised repository, available at `https://github.com/solid-state-survivor/ii-map`.

## 6 REPRODUCIBILITY

Due to the inherent randomness of our proposed training procedure, it is not possible to guarantee that a model trained from scratch would be identical to the model produced for this work. However, but we expect any such reproduction to yield sufficiently similar models and results. We provide a a streamlined set of instructions for use of our repository, described in Sec. 5, to train models as presented in Sec. 3.2.3, on how to obtain the required data, and how to use the model to produce all results presented in Figs. 2, 5, 6 and 7. We also provide our code for preparing the neural networks described in Sec. 2.1 for use in concept detection, in addition to the concept detection code itself. We additionally believe our code for training models as described in Sec. 3.2.3 to be sufficiently modular for reuse on other problem cases.

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

# A  APPENDIX

## A.1  INPUT STRUCTURE FOR TRAINED MODEL

Table 2: Input description of each input plane used for training our custom model with appended II-map module. Adapted for use from Hammersborg (2023).

| Plane number | Description |
| --- | --- |
| **0 − 5** | One plane for each piece-type for the player to move. (in the order of pawn, knight, bishop, rook, queen, king) |
| **6 − 11** | One plane for each piece-type for the opposing player. (in the order of pawn, knight, bishop, rook, queen, king) |
| **12 − 15** | Kingside, queenside castling rights for both players (not used in the presented variants) |
| **16** | If Black is the player to move. |
| **17** | Counter of the amount of moves since the last of capturing- or pawn-move. Used for the 50 move rule. (When no capturing- or pawn-moves have been made during the last 50 moves, any player can claim a draw.) |
| **18** | All zeros. |
| **19** | All ones. |

A.2    SUPPLEMENTARY CONCEPT DETECTION RESULTS

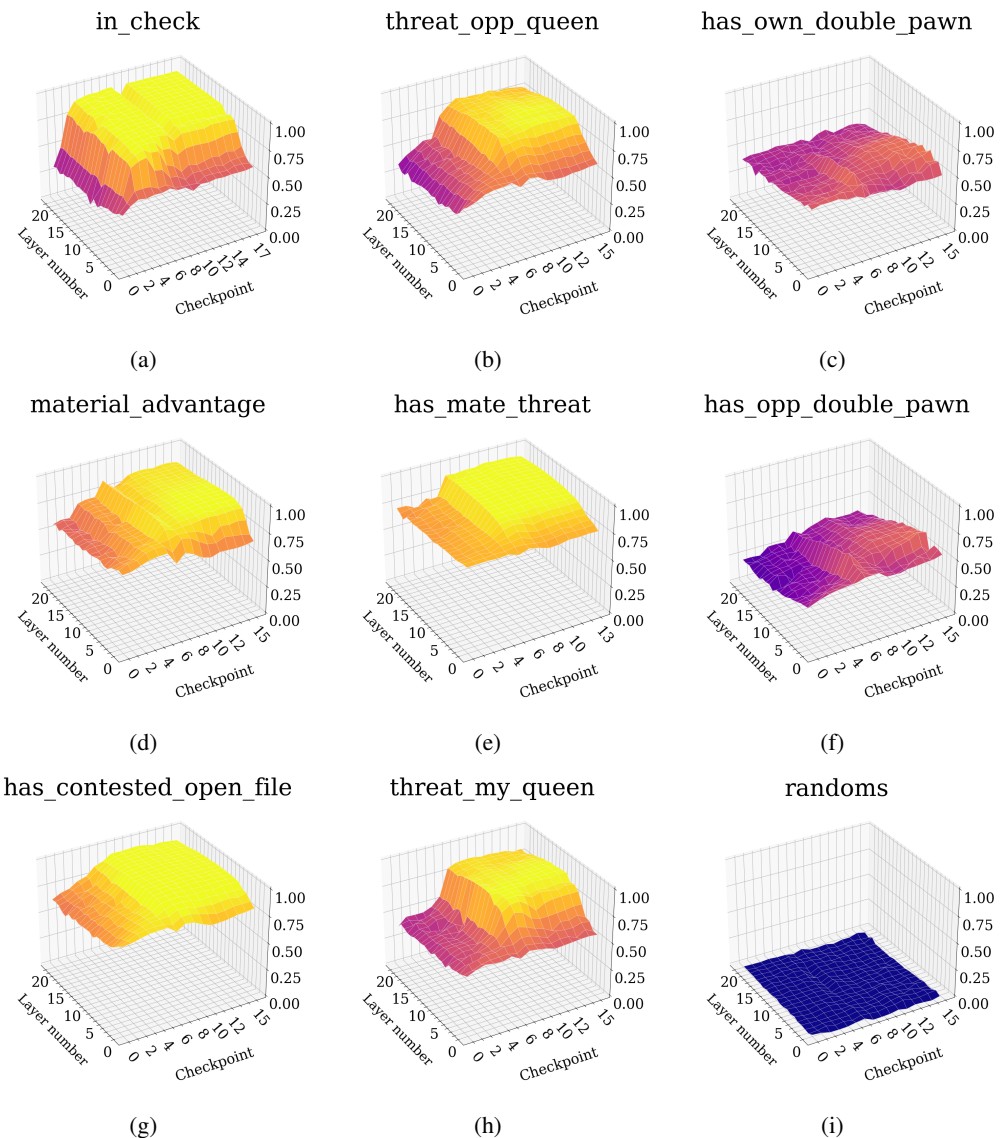

Figure 8: Concepts modelled by the 8x8 chess agent, showing the model's ability to detect (a) whether the player to move is in check, (b) whether the opponent's queen is under threat, (c) whether it has a double-pawn, (d) whether the player to move has a material advantage, (e) whether the opponent is currently presenting a mate-threat, (f) whether the opponent has a double-pawn, (g) whether both players contest an open file on the board, and (h) whether the player to move's queen is threatened, and (i) being a sanity check performed on a data set of random labels.

## A.3    Supplementary II-map results for various chess positions

Figure 9: Masks generated for a set of positions from well-known chess games for the models described in Sec. 3.2.3. The positions in Figs. (a) to (c) show positions from "The Game of the Century", Figs. (d) to (f) show positions from the first game between Garry Kasparov and IBM's DeepBlue, and Figs. (g) to (i) show positions from the sixth game from the 2021 World Championships.

