# OpenReview forum: "Information based explanation methods for deep learning agents -- with applications on large open-source chess models"
_ICLR.cc/2024/Conference — ICLR 2024 Conference Withdrawn Submission_

### Official Review · Reviewer_2i6y · 2023-10-29

**Soundness:** 1 poor
**Presentation:** 3 good
**Contribution:** 1 poor
**Rating:** 3
**Confidence:** 5

**Summary:**

The paper uses XAI methods for explaining factors that influence the moves of a chess engine. It claims two contributions:
1. to replicate the results of McGrath et al. with an open source chess engine
2. to propose a novel technique for computing salient squares.
The proposed technique is trained to learn a binary mask for the features along with the model, similar to an attention mechanism (which the authors do not relate to, btw).

**Strengths:**

The paper is well written and the approach is illustrated with good examples.
The confirmation of the results of McGrath et al. on LeelaChess data is also valuable, but I don't think that this is a research contribution for this conference. It might be more interesting for a computer games conference.

**Weaknesses:**

Of the two claimed contributions, I don't think that 1. is actually a contribution, as there is no reason to expect that results with LeelaChess would be any different than the reported results for AlphaZero. The second contribution - the new method - is maybe more interesting. However, it does not seem to stick out from related work, some of which the authors seem to be unaware of (in particular the work on SARFA, which was published 3 years ago at this conference (Piyust et al. 2020), but also follow-up papers such as (Fritz & Fürnkranz 2021).

The evidence for the validity of the proposed method is, as in previous work, only anecdotal, and I do not find the shown examples very convincing. The authors, e.g., claim that the examples shown in "...Fig. 5 capture the essence of what is necessary to solve the given puzzle....". Looking at these examples one sees that most (more than half) of the pieces are considered to be important in each of these positions, which includes pieces that are completely irrelevant for the assessment (e.g., Bd1 or Pb7 in Fig. 5 c)), and, on the other hand, miss pieces that are essential (such as Bd5 in Fig.5 c) which delivers the mate-in-1). The latter problem is briefly mentioned by the authors (but explained away), the former problem is not mentioned. Of course, if many pieces are annotated as important, there is a high chance that the essential pieces are among them. In more technical term, the proposed technique seems to have a high recall, but a rather low precision. Both of which are not evaluated.

I think the paper would need a somewhat more founded evaluation methodology. For example, an explanation could be considered useful if it helps a human chess player to find the best move in the position. I doubt that any of the shown explanations can actually do that. Prior works, such as the above-mentioned work on SARFA, tried to at least get a grip on the accuracy of the evaluation by comparing the method's salient squares with those mentioned by human annotators. I am not sure that these techniques work better than what is proposed here, but at least the make a stronger attempt in evaluating it.

Minor Comments
- The authors often use "author (year)" citations in places where "(author year)" is clearly required, which is often quite confusing, such as "Monte Carlo Tree Search Coulom" in the first sentence of 2.2.
- There seems to be an error in Figure 3 (the pawn on the f-file is moved backwards by a square).
- I did not understand the idea behind formula (4). Why the max over all possible pieces for unoccupied squares?
- The related work saliency-based and perturbation-based methods are not suitable for chess. I did not understand this. The authors themselves propose a form of saliency maps, so I think they do not refer to the representation but to how it is computed in the 5 given references, but they do not explain. For perturbations, they do give an explanation, but it does not seem to be convincing to me (in particular, as the above-mentioned SARFA is based on perturbations).


References:
G. Piyush, P. Nikaash, V. Sukriti, K. Dhruv, D. Shripad, K. Balaji, and S. Sameer, “Explain your move: Understanding agent actions using
specific and relevant feature attribution,” in International Conference on Learning Representations (ICLR), 2020.
Jessica Fritz, Johannes Fürnkranz: Some Chess-Specific Improvements for Perturbation-Based Saliency Maps. CoG 2021: 1-8

**Questions:**

Can you say something about the precision of the squares annotated? As mentioned above, occupied squares tend to be blue, others tend to be red, so that the pieces that are involved in the target move tend to be blue, but many others as well. Are these really helpful for understanding the position? (e.g., those mentioned above, or Pb7 in Fig. 5 b), etc.).

Your work also reminds me of attention mechanisms, maybe even transformer networks. Can you comment on that?

---

### Official Review · Reviewer_r57g · 2023-10-31

**Soundness:** 3 good
**Presentation:** 2 fair
**Contribution:** 2 fair
**Rating:** 3
**Confidence:** 2

**Summary:**

I will note that I have limited expertise in this type of RL research.

This work notes that the AlphaZero model used to play chess is not publicly available. Explainable AI (XAI) methods for concept detection were applied to the AlphaZero model and are also not public. This work utilizes a publicly available implementation of AlphaZero style implementation to play chess and applies XAI methods to it. The principal contribution is the use of concept detection with the publicly available chess agent implementation. The model employs a binary mask to mask out irrelevant information (tiles) in the training of the concept detection module. This mask signifies the pieces of most importance during the move. Results of the AlphaZero model are replicated using the open-source model, and numerous interpretability findings are shown and described in the context of the mask values.

**Strengths:**

- This is an important reproduction of an interpretability model applied to a longstanding RL problem.
- The method appears sound, and the experimental results are interesting. It is clear to see different concepts being identified through the mask values.

**Weaknesses:**

- This is a reproducibility paper with limited novelty. The methods are well-known. I don't think this work meets the bar to be accepted at ICLR.
- The results appear to be mostly qualitative, requiring the user to identify the meaning of different mask values in the context of the game positions.

I recommend rejection with weak confidence. I was able to understand the method but feel that I do not know enough about RL problems in the chess space. The paper still seems to lack novelty from my perspective

**Questions:**

N/A.  I would like to see other reviewer comments before asking additional questions.

---

### Official Review · Reviewer_8hN2 · 2023-11-01

**Soundness:** 2 fair
**Presentation:** 2 fair
**Contribution:** 3 good
**Rating:** 5
**Confidence:** 4

**Summary:**

This paper aims to be a re-implementation of Deepmind-Internal interpretability work on AlphaZero in Chess, as well as an introduction of a novel explainable AI method. The introduced method works by distilling the original model via imitation learning into a model with a special structure that ensures that only parts of the input are used, then use that structure to determine what is important for predicting the output of the original model.

**Strengths:**

The open-sourced implementation of interpretability for Alpha-zero should help others build on the work in the future. Explainability techniques that work for DeepRL policies are still a nascent field, so developing these on the most potent RL policies we have makes sense.

**Weaknesses:**

It is often unclear what in the paper is re-implementation and what is novel explainability work. Much of the paper is spent introducing their own technique, and eventually, they say that it achieves similar results as what was seen in the original paper, but it is unclear where the method they implemented deviates from prior work.  Given that part of the goal is the reimplementation of prior work, it would be ideal if the prior method were implemented to the extent that it re-produced existing results.

The proposed interpretability method is also several steps removed from the model itself, leaving a significant amount of room for error and interpretation. Rather than directly interpreting the model, the method clones the model into an architecture that would make it interpretable by construction. However, there is no guarantee that the resulting policy operate in the same way; we would expect the opposite. Moreover, if the instances we are trying to interpret are off-distribution for the cloned model, we cannot trust the results. The opposite is also true. If there is not enough diversity in the training of the cloned model, the model could also not need some information the original model does need.

There are also some systematic flaws with the objective of the cloned policy.  Since the policy doing the masking sees the whole board, there is information in the mask itself about the unseen pieces. For instance, the two networks could have a convention that if the second network does not see the king, it is in a typical place (as is pointed out in Figure 6). This back door makes it quite hard to draw a direct link between the parts of the board that are revealed and how the network works since we do not know what other information is implied by otherwise arbitrary choices in the mask.

**Questions:**

Is there a way to detect steganography between the two networks via the mask so it could at least be characterized?